# Accuracy and Precision of a Novel Photogate System to Measure Toe Clearance on Stairs

**DOI:** 10.3390/s23052429

**Published:** 2023-02-22

**Authors:** Timmion K. Skervin, Neil M. Thomas, Andrew J. Schofield, Mark A. Hollands, Constantinos N. Maganaris, Thomas D. O’Brien, Vasilios Baltzopoulos, Richard J. Foster

**Affiliations:** 1Research to Improve Stair Climbing Safety (RISCS), Faculty of Science, School of Sport and Exercise Sciences, Liverpool John Moores University, Byrom Street, Liverpool L3 3AF, UK; 2School of Psychology & Aston Research Centre for Healthy Ageing, College of Health and Life Sciences, Aston University, Birmingham B4 7ET, UK

**Keywords:** toe clearance, photogates, stair safety, optoelectronic

## Abstract

**Background:** Toe clearance on stairs is typically measured using optoelectronic systems, though these are often constrained to the laboratory, due to their complex setups. Here we measured stair toe clearance through a novel prototype photogate setup and compared this to optoelectronic measurements. **Methods:** Twelve participants (age 22 ± 3 years) completed 25 stair ascent trials, each on a seven-step staircase. Toe clearance over the fifth step edge was measured using Vicon and the photogates. Twenty-two photogates were created in rows through laser diodes and phototransistors. The height of the lowest photogate broken at step-edge crossing was used to determine photogate toe clearance. A limits of agreement analysis and Pearson’s correlation coefficient compared the accuracy, precision and relationship between systems. **Results:** We found a mean difference of −1.5 mm (accuracy) between the two measurement systems, with upper and lower limits (precision) of 10.7 mm and −13.8 mm, respectively. A strong positive correlation was also found (*r* = 70, *n* = 12, *p* = 0.009) between the systems. **Discussion:** The results suggest that photogates could be an option for measuring real-world stair toe clearances, where optoelectronic systems are not routinely used. Improvements to the design and measurement factors may help to improve the precision of the photogates.

## 1. Introduction

Insufficient toe clearance is one of the primary mechanisms for a fall over trip hazards such as stairs during daily activities. A stair fall can lead to severe injuries and serious consequences, particularly for older adults, such as hip fractures or a loss of independence [1]. The risk of tripping on stairs is typically assessed in laboratory environments by measuring the distance from the toe to a step edge during step crossing [2,3,4] and is referred to as toe clearance. Laboratory setups allow detailed measurements of toe clearance but may also impair natural stair walking. This may be due to the unnatural environment and the need to wear obtrusive retroreflective markers.

Optoelectronic systems are typically used in laboratories as the gold standard for assessing stair-walking behaviour and more generally human movement. These comprise multicamera systems that track light from either active or passive markers placed onto an individual to model body segments of interest and are accurate within millimetres [5]. To capture toe clearance, passive retroreflective markers can be placed on the shoe to define a foot segment, and the shoe tips [3] and/or plantar surfaces [6] of the shoe are then derived using virtual landmarks tracked by rigid marker clusters. The position of these virtual landmarks can be defined through digitising tools (such as marker probes/digitising wands [3,6]), a microscribe [2] or by tracing the border of the shoe [7,8]. For optoelectronic systems, multiple cameras are required for accurate measurement and specialist software is needed, meaning these systems are not very portable, require significant setup procedures and, additionally, are expensive to purchase. These constraints make real-world assessments of stair toe clearance challenging to perform with the optoelectronic approach. 

More portable alternatives for measuring toe/foot clearance include inertial measurement units (IMUs) [9], 2D video capture [10] and distance sensors placed on the soles of shoes [11]. Compared to optoelectronic systems, foot clearance measurements through IMUs and 2D cameras show errors of 7.4 mm [9] and 45 mm [10], respectively. Whilst the toe clearance errors with IMUs appear low, their application for real-world stair assessment may be limited through needing to secure instrumentation to the stair walker (which may become obtrusive and affect natural stair walking). Additionally, errors of 45 mm associated with 2D cameras may be too large to detect very low toe clearances or meaningful differences in the presence of stair modifications [12,13]. 

Photogates are characterised by the alignment of a light source to a respective sensor and natively act as a form of event detection once the line of light is intercepted, usually by an object or a person. Such systems are commonly used in applications for measuring time (often in sports performance [14]) but have also been utilised in basic physics experiments for the measurement of harmonic motion [15] through a programmable board. The advent of small powerful computer boards and microcontrollers, such as Raspberry Pi and Arduino, allow accessible and custom programming of electronic peripherals for diverse applications, including the control of light-based sensors allowing custom photogate setups. Such boards are small (approximately the size of a credit card), inexpensive and powerful. 

Here, we have developed and tested a prototype photogate setup, controlled through a Raspberry Pi computer for the application of measuring toe clearance on stairs. This photogate setup is portable, can be placed onto a step and uses inexpensive materials. Such a setup may be a useful tool in fall risk evaluations of real-world stair designs and/or stair modifications. In particular, as stair designs with inconsistently taller stair risers are known to result in reduced (less safe) toe clearances [8], visual cues on stairs can increase toe clearance [12]. Operation through a Raspberry Pi computer means the setup could remain on stairs for extended periods of time, allowing measurements from a large footfall of public stair users to inform stair safety evaluations.

We compare the accuracy and precision of our photogate setup to the gold standard optoelectronic approach to determine whether photogates could be a feasible option for real-world stair assessments of toe clearance. The aim of this study was to determine the precision and accuracy of a photogate setup against an optoelectronic system.

## 2. Materials and Methods

### 2.1. Participants

Twelve young adults (age 22 ± 3 years (range: 18–27 years), height 1.8 ± 0.08 m, mass 81.2 ± 19.30 kg) with no physical or neurological impairment were recruited from the University and local community and provided written informed consent to participate. To the authors’ knowledge, this preliminary study presents the first toe-clearance comparison of optoelectronic measurements to a prototype photogate setup meaning that no suitable prior data were available for sample size estimates, where expected mean differences and standard deviations are required [16]. A larger number of measurement pairs [17,18,19], however, were decided to moderate impacts of smaller sample sizes [20]. This study received ethical approval from the Institutional Review Board (17/SPS/002) and conformed to the Declaration of Helsinki.

### 2.2. Photogate Setup

Commercially available laser diodes (WayinTop, 5539) and phototransistors (Vishay TEPT4400) were used to create 22 photogates arranged one above the other on wooden blocks (Figure 1A). Each laser diode was manually aligned to its height respective phototransistor to create photogates approximately 1000 mm wide. Each photogate was vertically separated by approximately 6 ± 0.8 mm, and the first and last photogate were positioned 20 mm and 149 mm high, respectively (Figure 1B). These heights were used based on the toe clearance range found from our previous stair investigation [12], which all exceeded 20 mm and were below 149 mm. Laser diodes were powered by a 5 volts DC, mains-connected power supply and were connected in parallel. Phototransistors were connected to General Purpose Input/Output (GPIO) connections (maximum sampling rates of 125 MHz; Broadcom BCM2835 arm peripherals [21]) on a Raspberry Pi computer (Model 4, Raspberry Pi Foundation, Cambridge, UK, 2019) using pull-down resistors (10 kΩ). The Raspberry Pi’s 3.3 volts pin powered the phototransistors in a parallel circuit. The GPIO connections were programmed using Python (Python Software Foundation) to continually listen for the falling edge (high to low digital state change) that occurred when a photogate was broken. 

### 2.3. Validation 

Photogates were tested for validity by comparing the Vicon (Vicon MX, Oxford Metrics, UK, 2009; 26-camera motion capture system at 120 Hz) measured height of a 14 mm marker (attached to a rigid object) passed over a step edge, to the marker height measured concurrently by photogates abutted to the step edge over 150 trials. The marker was affixed to the bottom of a rigid object and was passed linearly over the step edge in an anterior direction, such that the marker would break the photogates first as opposed to the rigid object (Figure 1A). 

### 2.4. Stair Ascent Trials

Participants completed 25 trials ascending a seven-step staircase (rise heights of 200 mm and tread lengths of 250 mm) at a self-selected speed. Trials began approximately two/three steps away from the staircase from the same starting position. Participants ascended the stairs in a step-over-step manner, crossing the first step with the same self-selected foot for each trial. Toe clearance of the lead foot over a single step edge (step 5) was captured using Vicon and the photogates (abutted to step edge) concurrently. 

### 2.5. Rigid Object and Toe Clearance Measurement

A cluster of three retroreflective markers were placed on the dorsal surface of the mid-forefoot for tracking a shoe-tip virtual landmark. A digitising wand (C-Motion, Germantown, MD, USA, 2016) was used to establish a virtual landmark on participants’ shoe tips (the most anterior and inferior aspect of the shoe) and the location of step edge 5. Vicon-measured marker height/toe clearance was defined as the vertical distance of the marker/shoe-tip landmark to step edge 5. These measures were extracted at the point, where the difference in anterior/posterior position between step edge 5 and the marker/shoe-tip landmark was zero. Marker data were labelled and gap-filled in Vicon, filtered using a fourth-order Butterworth bidirectional filter (cut-off frequency 6 Hz) and analysed using Visual 3D (C-Motion, Germantown, MD, USA, 2022). Photogate measurements of marker height were extracted as the lowest photogate broken. Toe clearance was measured as the lowest photogate broken within a time window of 8.3 ms from the first photogate breaking to isolate measurements only to the period when the lead toe passes the step edge. All photogate data were analysed using Python. 

### 2.6. Statistical Analysis

Residual plots were used to visually inspect the data for normality. A limits of agreement analysis [22] was performed on the marker height and toe clearance data. Such analysis determines the mean difference between the two measurement methods (bias/accuracy), along with 95% agreement limits that determine the precision (range of agreement). Individual trials were used for the comparison of marker height, and participant averages were used for the comparison of toe clearance, resulting in 12 data pairs. Data were averaged for the toe clearance comparison as the limits of agreement method assumes independent observations [23]. Bland–Altman plots were created to assess how close the photogate measurements agreed with the Vicon measurements for the marker height and toe clearance trials. Pearson’s correlation was used to assess the relationship between the photogates and Vicon for the marker height and toe clearance trials. Statistical analyses were conducted using R [24] and the BlandAltmanLeh software package [25].

## 3. Results

### 3.1. Rigid Object Comparison

For the marker height measurements, the limits of agreement analysis revealed a mean difference (accuracy) of −1.4 mm (photogates overestimated marker height) between the two measurement systems, with upper and lower limits (precision) of 3.5 mm and −6.4 mm, respectively (Figure 2). The mean and standard deviation for the photogates and Vicon measurements were 72.7 ± 33.8 mm (range: 20–142.8 mm) and 71.3 ± 34.2 mm (range: 17.7–142.1 mm), respectively. A very strong positive correlation was found for the marker height measurements between the photogates and Vicon (*r* = 99, *n* = 150, *p* < 0.0001). 

### 3.2. Toe Clearance Comparison

A total of 296 trials were included for the toe clearance comparison (data were missing from four trials due to incomplete recordings). The limits of agreement analysis revealed a mean difference of −1.5 mm (photogates overestimated toe clearance) between the two measurement systems, with upper and lower limits of 10.7 mm and −13.8 mm, respectively (Figure 3). The mean and standard deviation for the photogate and Vicon measurements were 53.1 ± 7.7 mm (range: 44.6–68 mm) and 51.5 ± 8.5 mm (range: 30.8–61.3 mm), respectively. A strong positive correlation was found between the photogates and Vicon toe clearances (*r* = 70, *n* = 12, *p* = 0.009). 

## 4. Discussion

This report demonstrates the first use of a novel prototype photogate setup that shows accuracy and a strong positive correlation when compared to an optoelectronic system for measuring vertical toe clearances over a single step during stair ascent. 

The photogates were found to accurately measure toe clearance and correlate strongly with the Vicon measurements, albeit resulting in a wider range of agreement when compared to measurements of marker height. The Vicon system was chosen as the criterion method due to the high level of precision (millimetre accuracy) offered through these systems. However, differences in agreement between the rigid object comparison and stair assessment indicate that sources of error in toe clearance agreement may relate to the Vicon approach in quantifying toe clearance through a virtual landmark. The ground truth for toe clearance was based on the Vicon measurement of a single shoe-tip virtual landmark at step-edge crossing. However, the first photogates broken may not have always been caused by the aspect of the shoe, where the virtual landmark was located (the most anterior and inferior aspect of the shoe) due to foot orientation at step-edge crossing. This could have led to measurement discrepancies between the two systems as the photogate measurements were based on the lowest photogate broken. Assessing this would require multiple virtual landmarks covering a larger shoe surface area [6]. This would help in identifying the shoe aspect that crosses the step edge first for the measurements of toe clearance through Vicon and could lead to a better agreement range between the two systems. Importantly, the marker height results support our photogates as a setup for measurements of height over a step edge. The agreement in measured marker height might be strengthened by using smaller markers. Vicon uses the marker centroid for positional tracking, whilst the photogates measure marker height based on the lowest photogate broken. The difference between the point intercepting the photogate on the marker and the centroid reflects a small and inherent measurement offset between systems, but this margin may be minimised with smaller markers and is a limitation within this comparison.

Improving the prototype version of our photogates may require increasing the photogate spatial resolution or further controlling the state change (indicated by the falling edge detection) on the GPIO connections with analogue to digital converters. Increasing the spatial resolution will allow for a more precise measurement and may be helpful when small differences in foot clearance may be of interest. This study used a resolution of ~6 mm for the photogates, whilst optoelectronic systems are capable of resolutions of ~1 mm [26]. Increases to the spatial resolution could be achieved by reducing the housing on the components of the photogates (such as the laser diodes), though clearly there will be a space limit on how many photogates can ultimately be set up and correctly aligned. The state change on the GPIO connections operates on the level of voltage surpassing a threshold and the voltage changes as a function of light intensity on the phototransistor. The Raspberry Pi board cannot natively read analogue signals directly, meaning it was not clear whether some phototransistors were closer to a state change than others based on having more/less light. This means for some phototransistors more/less of the light would need to be broken for a change to be registered and might explain the presence of the small number of outliers found in the rigid object comparison. This could be resolved by adding analogue to digital converters to each phototransistor channel, which would provide voltage values and help indicate how close each phototransistor was to this threshold.

The −1.5 mm mean difference between measurement systems suggests that photogates can potentially be used to measure toe clearance on real-world stairs, overcoming the setup/operation complexities associated with optoelectronic systems and the requirement of attaching IMUs to a stair walker. Compared to IMUs and 2D cameras, our photogate setup shows greater accuracy (IMUs 7.4 mm error [9]; 2D cameras 45 mm error [10]), though measurements in this study were performed over a single step. Measurements over multiple steps could be performed by repeating the photogate setup, though, clearly, this will increase the amount of materials and power required for operation. The height of our photogates in this study covered a range of typical foot clearances known to us based on our previous investigation involving young and older adults [12]. Although the primary aim of this study was to determine the validity of a prototype photogate setup for measuring toe clearance, we acknowledge that toe clearances below 20 mm do occur [2] and it is a limitation of the measurement range chosen. Accounting for measurements below 20 mm should be considered in future design iterations for real-world applications. 

A limitation of the photogate system involves manually affixing and aligning the photogates which can be time-consuming. This may be overcome through permanent fixings and 3D-printed enclosures. The existing design of the device is also cumbersome and distracting to stair users. Converting the photogates into a professional, robust and compact setup that can be placed repeatedly on public stairs for prolonged time periods is desirable. The system could be further adapted to measure very low toe clearances (i.e., below 20 mm) and horizontal toe clearance during stair descent. The application of our photogate setup may also translate in principle to other trip/fall hazards, such as raised surfaces or obstacles, where sufficient toe clearance is similarly required for safe crossing. The photogates would be particularly useful for measuring the impact of previously developed stair illusions on toe clearance [3,12].

## 5. Conclusions

This investigation shows that our novel prototype photogate setup is accurate for measuring stair toe clearances. Improvements to the agreement range may require a more detailed approach with shoe-tip virtual landmarks and photogate design factors. Addressing these limitations will help convert the current prototype into a complete system that can provide an alternative to optoelectronic systems for measuring toe clearances outside the laboratory environment. 

## Figures and Tables

**Figure 1 sensors-23-02429-f001:**
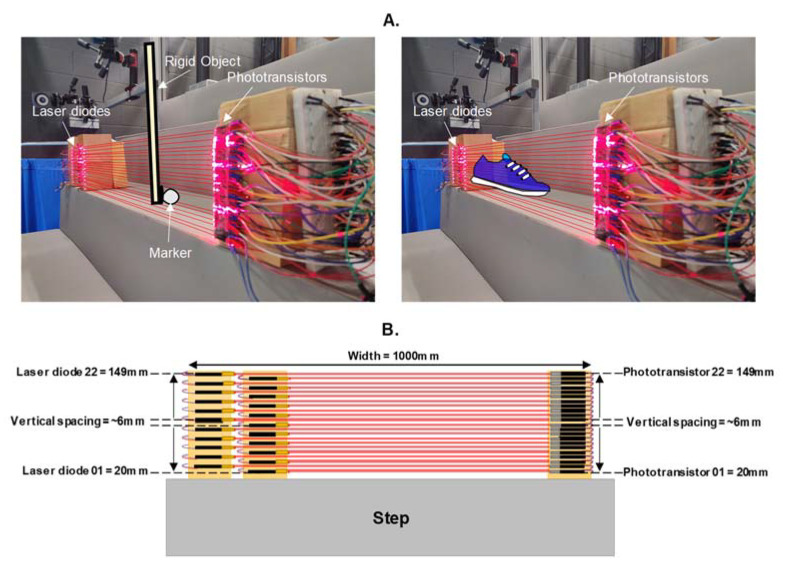
(**A**). A photogate setup using laser diodes and phototransistors for the measurement of marker height/toe clearance. Two blocks were used for affixing the laser diodes to allow space above and below for fine adjustments with photogate alignment. The height of the lowest photogate broken by the marker/shoe toe tip was used as the marker height/toe clearance (seventh photogate in the annotation, measuring 59.4 mm in height). (**B**). A frontal plane illustration of the photogate dimensions and setup.

**Figure 2 sensors-23-02429-f002:**
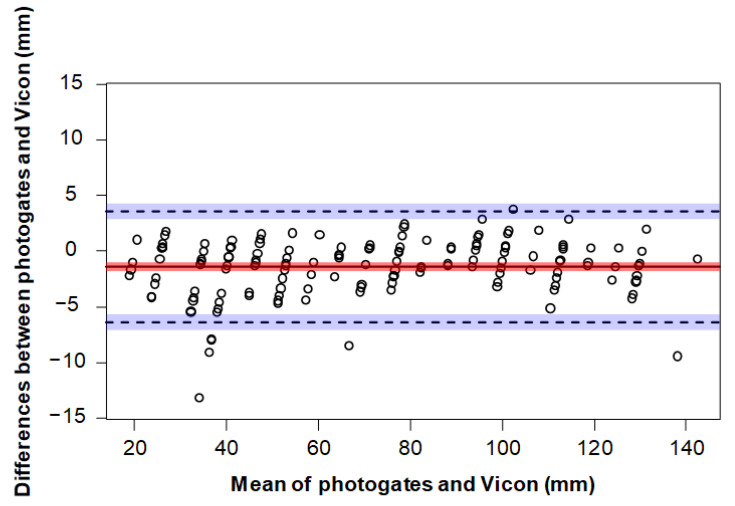
A Bland–Altman plot representing the mean (x axis) and differences (y axis) of a marker height measured by Vicon and the photogates. The limits of agreement are indicated as dotted lines with 95% confidence intervals (blue shading). The bias is represented as a solid line with 95% confidence intervals (red shading).

**Figure 3 sensors-23-02429-f003:**
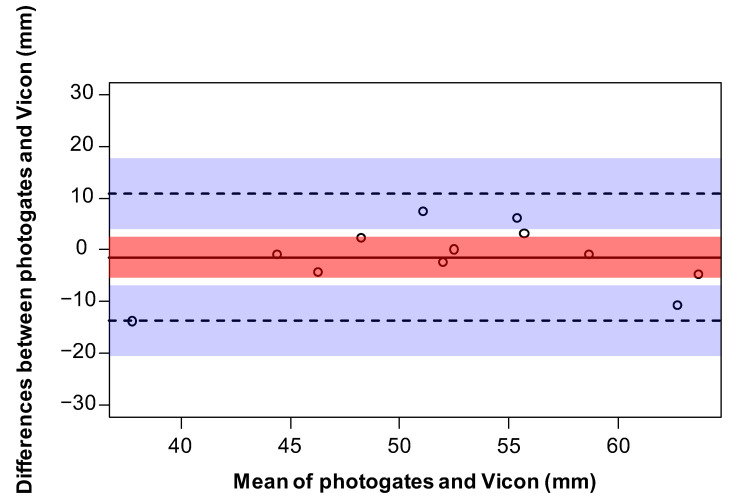
A Bland–Altman plot representing the mean (x axis) and differences (y axis) of toe clearances measured by Vicon and the photogates. The limits of agreement are indicated as dotted lines with 95% confidence intervals (blue shading). The bias is represented as a solid line with 95% confidence intervals (red shading).

## Data Availability

The data presented in this study are available as Appendix A.

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
