# Peer review of "Accuracy and Precision of a Novel Photogate System to Measure Toe Clearance on Stairs"

_sensors, 2023, doi:10.3390/s23052429_

Round 1

Reviewer 1 Report

The article presents a novel photogate setup prototype for measuring toe clearances on stairs. Authors performed experiments with the developed prototype, and carried out comparisons and validation through a Vicon device. Number of trials conducted in all experimentations seems to be statistically adequate. It is a concise article, well-structured and very well-written. Authors are recommended to address the following comments.

- Is there a specific reason for the use of cm's throughout the text? As a unit, mm's seem to be a more appropriate choice when magnitudes of measurements are investigated.

- Authors are recommended to elaborate on why measuring toe clearances is essential and photogates are a good solution for this. How can the collected data be utilised in stair designs and structures or for any other means?

Author Response

Thank you for your feedback. Please see our responses in the attachment. 

Rich 

Reviewer 2 Report

Title:  Accuracy and Precision of a Novel Photogate System to Measure Toe Clearance on Stairs

In this study, the authors investigated the new measurement system that toe clearance on stairs. With all humility, these recommendations are collected with the intention that they can help to improve this work.

-          Toe clearance is a very important part of fall prevention. However, falls usually occur on flat land, on land with obstacles, or on crowded streets, so why did you choose to walk on stairs?
How do you plan to apply the results of this study clinically?

-          Please add IRB approval authority and number to 2.1 Participants.

-          Describe the method and procedure for extracting the sample size to verify the effectiveness of this study.

-          Line 97: Pi computer (Model 4, Raspberry Pi Foundation, UK). > Add year of manufacture/ Equipment specifications: model name, company name, country of manufacture, year of manufacture

-          Line 104: Vicon (Vicon MX, Oxford Metrics, UK). > Add year

-          Line 128: Visual 3D (C-Motion, Germantown, MD, USA). > Add year

-          Why did you choose the seven-step stage case in the step walk of the experimental method of this study?

-          Toe clearance for fall prevention is considered an important study to improve the walking ability of the elderly and stroke patients.

Author Response

(The authors gave the same response as above.)
